# Potential Roles for the GluN2D NMDA Receptor Subunit in Schizophrenia

**DOI:** 10.3390/ijms241411835

**Published:** 2023-07-23

**Authors:** Chitra Vinnakota, Matthew R. Hudson, Nigel C. Jones, Suresh Sundram, Rachel A. Hill

**Affiliations:** 1Department of Psychiatry, School of Clinical Sciences, Faculty of Medical, Nursing and Health Sciences, Monash University, Clayton, VIC 3168, Australia; chitra.vinnakota@monash.edu (C.V.); suresh.sundram@monash.edu (S.S.); 2Department of Neuroscience, Faculty of Medical, Nursing and Health Sciences, Monash University, Melbourne, VIC 3004, Australia; matt.hudson@monash.edu (M.R.H.); nigel.jones@monash.edu (N.C.J.); 3Mental Health Program, Monash Health, Clayton, VIC 3168, Australia

**Keywords:** GluN2D, schizophrenia, NMDA receptor, NMDAR antagonists

## Abstract

Glutamate *N*-methyl-D-aspartate receptor (NMDAR) hypofunction has been proposed to underlie schizophrenia symptoms. This theory arose from the observation that administration of NMDAR antagonists, which are compounds that inhibit NMDAR activity, reproduces behavioural and molecular schizophrenia-like phenotypes, including hallucinations, delusions and cognitive impairments in healthy humans and animal models. However, the role of specific NMDAR subunits in these schizophrenia-relevant phenotypes is largely unknown. Mounting evidence implicates the GluN2D subunit of NMDAR in some of these symptoms and pathology. Firstly, genetic and post-mortem studies show changes in the GluN2D subunit in people with schizophrenia. Secondly, the psychosis-inducing effects of NMDAR antagonists are blunted in GluN2D-knockout mice, suggesting that the GluN2D subunit mediates NMDAR-antagonist-induced psychotomimetic effects. Thirdly, in the mature brain, the GluN2D subunit is relatively enriched in parvalbumin (PV)-containing interneurons, a cell type hypothesized to underlie the cognitive symptoms of schizophrenia. Lastly, the GluN2D subunit is widely and abundantly expressed early in development, which could be of importance considering schizophrenia is a disorder that has its origins in early neurodevelopment. The limitations of currently available therapies warrant further research into novel therapeutic targets such as the GluN2D subunit, which may help us better understand underlying disease mechanisms and develop novel and more effective treatment options.

## 1. Schizophrenia—An Overview

Schizophrenia is a severe, debilitating, chronic neuropsychiatric disorder with a lifetime prevalence of 0.72% [1]. It has a complex and heterogeneous presentation making it difficult to diagnose and to identify a consistent underlying aetiology or pathology. People with schizophrenia demonstrate high inter-individual variability with respect to symptoms, disease course, outcome and response to treatment. Schizophrenia symptoms are divided into three main classes: positive, negative and cognitive deficits. Of the three, positive symptoms are the most easily identifiable and are defined as psychotic features that are not usually present in healthy people, and include hallucinations, delusions and disorganised speech and behaviour [2,3]. Negative symptoms refer to a reduction or disruption in normal emotions and behaviours manifesting as social and emotional withdrawal, apathy and avolition [2,4]. Cognitive symptoms vary in severity amongst people with schizophrenia and include elements such as deficits in verbal memory, working memory, attention, executive functioning, cognitive flexibility and processing speed [5].

Schizophrenia typically has an early onset, with most people being diagnosed in their late adolescence—early adulthood years. The onset of the first psychotic episode is usually preceded by a prodromal period during which symptoms gradually emerge and this period can last several years. An early onset combined with long-term deficits in social, educational and occupational function make this disorder one of the leading causes of chronic disability with significant impacts on the quality of life of patients and their families and caregivers [6,7]. Additionally, this disorder places a substantial economic burden on health care systems with estimates of annual associated costs reaching up to USD 150 billion in the United States and AUD 4.9 billion in Australia [8,9].

The causes of schizophrenia are not fully understood but are thought to be multifactorial, involving a complex interplay between multiple genetic variants and environmental factors. Genome-wide association studies have identified multiple common variants of small effect spanning over 250 genetic loci, suggesting that schizophrenia is a polygenic disorder in most cases [10,11]. Genes associated with schizophrenia risk are involved in various functions, including the regulation of the postsynaptic membrane, synaptic transmission and neurodevelopmental pathways, including glutamate pathways [11,12,13]. Non-genetic factors that increase lifetime risk for schizophrenia include obstetric complications, advanced paternal age, living in an urban setting, childhood trauma or adversity, cannabis use and first-generation migration [14,15,16,17,18,19,20,21]. Although there have been advances in our understanding of risk factors associated with schizophrenia, the aetiological complexity has made it a challenge to identify the underlying disease mechanisms and find effective cures and preventative strategies.

Currently available pharmacological treatments, chiefly conventional and atypical anti-psychotics, and psychotherapy, have proven clinical utility and can help manage positive symptoms in some people [22,23]. However, antipsychotic medications have limited efficacy and are poorly tolerated, with substantial side effects in approximately 30% of people. Moreover, they usually offer little benefit in improving negative and cognitive symptoms, and these symptom types therefore remain a pressing, unmet medical need [22,24]. The prevalence, burden and current lack of effective treatments for schizophrenia highlight a need to improve our understanding of the underlying mechanisms and neurobiology of the disorder in order to identify and develop better treatments.

Although the predominant literature has focused on dopaminergic dysfunction in schizophrenia [25,26,27,28], and many of the currently available anti-psychotics target dopaminergic receptors [29,30,31,32], the limitations of these treatments suggests that other pathological processes may be involved. A theory gaining increasing support for the aetiology of schizophrenia is the hypofunction of one of the major glutamate receptor subtypes, the *N*-methyl-D-aspartate receptor (NMDAR) [33,34,35,36].

## 2. Glutamatergic Signalling in the Central Nervous System

Glutamate plays a key role in mediating the homeostatic balance between excitation and inhibition in the brain, cortico-cortical neurotransmission, neuronal development, neurodegeneration, nervous system plasticity and learning and memory [37]. Glutamate carries out its actions through its receptors which are divided into two groups: the ionotropic and metabotropic receptors [38,39]. The ionotropic receptors, namely the NMDA, α-amino-3-hydroxy-5-methyl-4-isoxazolepropionic acid (AMPA) and kainate receptors, are integral membrane proteins composed of four large subunits (>900 residues) and act as ligand-gated cation channels [40]. The metabotropic receptors are G-protein-coupled and activate intracellular biochemical cascades [39,41]. As this review is focused on NMDARs, specifically the GluN2D subunit and its relevance to schizophrenia, we direct readers to the following review articles [42,43,44] for further discussion of the role of AMPA, kainate and metabotropic receptors in schizophrenia.

### NMDA Receptor Structure and Function

NMDARs are widely distributed and can be found on both neuronal and non-neuronal cells. These receptors play a key role in many physiological processes, including neurodevelopment, synaptogenesis and locomotion, and due to their role as critical mediators of activity-dependent synaptic plasticity, they are especially important for learning, memory formation and other forms of cognition [45,46,47,48,49,50,51]. NMDARs are unique in that they require the concomitant binding of glycine and glutamate along with membrane depolarisation for activation and are thus referred to as ‘molecular coincidence detectors’ [52,53,54]. At resting membrane potential, extracellular magnesium (Mg2+) can be found in the ion channel pore, blocking NMDARs in a voltage-dependent manner. Partial depolarisation of the neuron relieves the blockade, opening the channel. The subsequent Ca2+ influx into the neuron triggers a cascade of events that can influence both local, acute functional synaptic plasticity and, via changes in gene expression, sustained neural plasticity [55].

The heteromeric composition of NMDARs enables different pharmacological, biophysical and functional properties for the receptor, and these heteromers vary in distribution and expression, both regionally and temporally, throughout development. The different subunits that make up the NMDAR are termed GluN1, GluN2 and GluN3. The GluN1 subunit is encoded by a single gene but has eight splice variants [56]. There are four different GluN2 subunits (A–D), encoded by four different genes and two different Glun3 subunits (A and B) [39,57]. A functional NMDAR is typically composed of two GluN1 subunits and two subunits from among the GluN2A-D and GluN3A-B subunits [45,58]. The obligatory GluN1 subunit is ubiquitously expressed throughout the brain and over the lifespan [39,59]. Of the GluN2 subunits, which have a more varied and complex temporal and spatial expression, GluN2A and GluN2B are the predominant subtypes found in the adult human brain, whilst the GluN2C and GluN2D subunits are more highly expressed in the developing brain [45,60]. The subunit composition of NMDARs influences its functional properties, including agonist affinity, Mg^2+^ block, decay kinetics and modulation by polyamines [61,62]. Given the importance of NMDAR subunits in mediating normal brain function, it is not surprising that the dysfunction of these subunits has been linked to various neurological diseases, including schizophrenia. The NMDAR hypofunction model is one of the most commonly adopted and supported models of schizophrenia and is often employed to study the aetiology and pathology of the disorder as well as for the development of novel treatment strategies.

## 3. NMDA Receptor Hypothesis of Schizophrenia

From the late 1950s, phencyclidine (PCP) and ketamine have been reported to induce positive, negative and cognitive phenotypes such as psychosis-like dissociative states and neurocognitive disturbances in healthy individuals like those observed in people with schizophrenia [63,64,65,66,67,68]. Furthermore, these drugs exacerbate symptoms, including psychosis in individuals with schizophrenia [66,69,70]. The NMDAR hypothesis was first proposed soon after, in the late 1970s–1980s, when it was found that PCP and ketamine carry out their actions via NMDAR blockade [64,71,72,73]. The extensive literature drawing parallels between NMDAR antagonism and many of the unique features of schizophrenia have led some to postulate that these antagonists might act via the same mechanisms that become dysfunctional in this disorder [74,75,76]. In support of this, studies in rodents have shown that knockdown of the obligatory GluN1 subunit of the NMDAR can result in phenotypes relevant to schizophrenia, including hyperlocomotion (proposed to be indicative of a striatal hyperdopaminergic state which underlies positive symptoms such as psychosis), anhedonia, social deficits, impaired learning and abnormal neuronal oscillations [77,78,79,80,81,82].

Although NMDAR hypofunction has been linked to schizophrenia symptoms, the precise underlying mechanisms are still unclear. One hypothesis is that it is primarily the dysfunction of NMDARs on GABAergic interneurons, rather than more widespread NMDAR dysfunction, which contributes to the molecular, physiological and behavioural characteristics of schizophrenia [35,83]. GABAergic interneurons are stimulated by postsynaptic NMDAR activation and, in turn, synapse onto excitatory glutamatergic pyramidal cells in a negative feedback loop. GABAergic interneurons connect to hundreds of pyramidal cells in this manner, enabling them to coordinate synchronised network activity throughout the brain, including the hippocampus. The activity of these glutamatergic pyramidal cells, in turn, drives downstream striatal dopaminergic neurons (Figure 1) [84,85]. In this way, any disruption to normal NMDAR function would lead to downstream GABAergic and dopaminergic dysregulation. In line with this theory, studies conducted in healthy humans and in rodent models show that administration of NMDAR antagonists counterintuitively results in the hyperactivity of cortical pyramidal neurons and enhanced cortical glutamate release by disinhibition [86,87,88,89,90,91,92,93,94,95]. This is corroborated by brain imaging data in both humans and rodents showing that the administration of an NMDAR antagonist results in net excitation and decreased coordinated burst-firing in the prefrontal cortex, which is linked to cognitive deficits [89,96,97,98,99]. This cortical disinhibition suggests that NMDAR antagonists may be preferentially targeting the NMDARs on GABAergic interneurons. Additionally, NMDAR antagonists have been shown to reproduce schizophrenia-like dopaminergic dysfunction, including an increase in striatal D2 receptors and dopamine release [100,101,102,103]. This striatal hyperdopaminergic state has been linked to the positive symptoms of schizophrenia. In this way, NMDAR hypofunction might be resulting in a cortical excitatory/inhibitory (E/I) imbalance, affecting the ability of interneurons to synchronise firing across neural networks, and driving the changes in the dopaminergic system, ultimately culminating in the clinical symptoms of schizophrenia.

In recent years, there has been mounting evidence of the dysfunction and/or dysregulation of NMDAR-mediated neurotransmission in schizophrenia. Several schizophrenia candidate genes that affect NMDAR signalling have been identified, including *GRIN1*, *GRIN2A*, *GRIN2B* and *GRIN2D*, which are genes encoding the GluN1, GluN2A, GluN2B and GluN2D subunits of the NMDAR, respectively, as well as the gene encoding D-amino acid oxidase (DAAO), which regulates the availability of D-serine, a co-agonist at NMDARs [11,104,105,106]. Findings from other genetic and post-mortem patient studies have further suggested that NMDAR signalling is disrupted in schizophrenia [33,107,108,109].

The contribution of the GluN1, GluN2A and GluN2B subunits to schizophrenia pathology and symptoms has already been well researched and reported (see [34,110,111,112]). This review will instead focus on the relatively understudied GluN2D subunit, as its unique developmental expression pattern, cellular specificity and electrophysiological properties suggest that it may play a role in the NMDAR hypothesis of schizophrenia. To this end, we firstly outline some of these unique physiological features, and then describe the changes observed in this subunit in people with schizophrenia. Finally, we discuss how alterations in the expression and function of the GluN2D subunit may contribute to schizophrenia.

## 4. GluN2D Subunit

### 4.1. GluN2D Receptor Subunit Expression and Distribution

The GluN2D (common names: NMDAR2D, NR2D, GluRε4) subunit is encoded by the *GRIN2D* gene, which consists of 13 exons, spanning 49.3kB, and is located on chromosome 19q13.1-qter in the human genome [39]. The *GRIN2D* gene has two potential splice isoforms (NR2D-1 and NR2D-2), the longest of which contains 1356 amino acids [39,113]. In rodents, GluN2D has been extensively characterized; expression levels are first detected between embryonic day (E)15 and 18 during late-embryogenesis, with levels peaking by post-natal day (P)7–10 [114,115]. The GluN2D mRNA and protein expression levels decrease gradually after the early neonatal phase until late adolescence (P40–50), when they reach their relatively low steady-state expression level [61,114,115,116]. During the embryonic and early neonatal phases, the expression of GluN2D is widespread and detected in several regions, including the spinal cord, midbrain nuclei, diencephalon (thalamus, hypothalamus), certain basal ganglia nuclei (substantia nigra and subthalamic nucleus), retina, olfactory bulb, cerebellum, cerebral cortex and hippocampus [60,61,113,114,117]. The ubiquitous distribution of the GluN2D subunit during the early phases of life makes this subunit particularly interesting amongst the NMDAR subunits as it suggests that GluN2D plays a critical role in modulating circuit connectivity and function during neurodevelopment. This could be of significance when considering a disorder such as schizophrenia that is thought to have its origins in early development [118,119].

As the rodent ages and GluN2D expression levels reduce, it becomes more localized to distinct cell subtypes [61,114,115]. This is especially apparent in the hippocampus and cortex, where multiple studies have shown that GluN2D is enriched in parvalbumin (PV)-containing GABAergic interneurons in mature rodents, whereas its expression and activity on glutamatergic pyramidal cells decreases [117,120,121,122,123]. Electrophysiological analyses from the adult mouse medial prefrontal cortex (mPFC) showed that a GluN2C/D positive allosteric modulator, CIQ(+), increased the intrinsic excitability of interneurons and enhanced excitatory postsynaptic currents (EPSCs) from interneurons, whilst not having any effect on the surrounding pyramidal cells [121]. Another study found that while CIQ(+) enhanced EPSCs in WT mice, this effect was missing in GluN2D KO mice [122]. The sustained expression of GluN2D-containing receptors on interneurons over development may suggest that this subunit is required to maintain proper inhibitory drive and control overall circuit function [124,125].

In humans, the spatial and temporal expression of GluN2D is thought to be consistent with that reported in rodents; however, it is yet to be well characterised. In human foetal brains, GluN2D mRNA is abundantly expressed and is one of the predominant NMDAR subunits expressed between gestational weeks 8 and 20 [126]. In contrast, a study of neurologically normal, adult human post-mortem brains reported only moderate expression of GluN2D mRNA in the prefrontal, parietal and motor cortices, where instead the major subtypes expressed were the GluN2A and GluN2B subunits [127]. This study also reported that whilst GluN2D expression was low within most neurons in the hippocampus, expression was moderately intense within a small subset of hippocampal neurons, particularly in the hilus, a region containing many interneurons [127]. A recent study which used laser microdissection to isolate pyramidal neurons and PV+ interneurons from the human post-mortem dorsolateral PFC (dlPFC), combined with RNA sequencing and microarray, provided evidence that the GluN2D subunit is indeed particularly enriched in PV+ interneurons, but not pyramidal neurons [121]. It should be noted, however, that whilst the interneurons that were isolated for this study expressed markers specific to PV+ interneurons, such as KCNS3, they also expressed markers common to other interneuron types, such as somatostatin, and thus might not be delimited to PV+ interneurons alone [121]. Another post-mortem in situ hybridization study revealed that in control human brains, the GluN2D subunit is preferentially expressed in layers II, IIIa-c, IV, Va and VIa of the cerebral cortex and at much higher levels in the prefrontal cortex compared with the parieto-temporal or cerebellar cortices [128].

Multiple studies examining the subunit composition of NMDARs in the central nervous system suggest that while the majority of GluN2D subunits are associated with the GluN1 subunit, the GluN2D subunit also forms heteromeric assemblies with GluN2A and/or GluN2B subunits in different brain regions and neuronal subpopulations [129,130,131]. For example, the GluN2D subunit is shown to primarily form a binary complex with GluN1 in the thalamus of young (post-natal day 7) and adult rats, whereas this binary complex is not detected at all in the rat midbrain, where instead the GluN2D subunit forms triheteromeric assemblies with GluN1 and GluN2A or GluN2B [129]. The varying spatial expression of the different subtypes of GluN2D-containing NMDARs suggest distinct functional roles.

### 4.2. GluN2D Receptor Subunit Function

Glutamate displays 5–6 times greater potency at GluN2D-containing NMDARs than GluN2A- or GluN2B-containing NMDARs [39,132]. Similarly, GluN1 agonists such as glycine are most potent when the GluN2 subunit in the NMDAR is GluN2D [133]. GluN2D-contaning NMDARs also have a weak Mg2+ block that is 10-fold lower than that of GluN2A- or GluN2B-containing receptors and are also reported to have a slightly lower Ca2+ permeability [134,135]. The resistance to Mg2+ block suggests that neurons expressing the GluN2D subunit may be more responsive to synaptic glutamate release. The non-competitive NMDAR antagonist ketamine is more potent at, and shows approximately five-fold selectivity for GluN2D-containing receptors, compared with GluN2A or GluN2B subunits [136]. This, combined with the fact that GluN2D-containing receptors are predominantly expressed on interneurons, may suggest that the GluN2D-containing receptors might be involved in the cortical disinhibition induced by certain NMDAR antagonists such as ketamine [121,137]. Another unique feature of GluN2D-containing receptors is that they have the slowest deactivation rate of the GluN2 receptor subtypes, exhibiting 10-fold slower receptor deactivation than GluN2B and GluN2C receptors and 100-fold slower than GluN2A receptors [39,62,138]. Application of glutamate to NMDARs containing the GluN2A subunit was found to generate a macroscopic current with a deactivation time of tens of milliseconds compared with several seconds for GluN2D-containing NMDARs [138]. Importantly, this prolonged decay time is matched with an almost complete lack of desensitization, giving this subunit the capacity to shuttle large quantities of ions across the plasma membrane in order to sustain depolarization for a long enough period to initiate burst-firing [61,138,139]. This ability to generate burst-firing would be especially important for the interneurons on which they are expressed, possibly at least partially contributing to their rhythmic, oscillatory nature [137]. Thus, the unique electrophysiological properties of the GluN2D subunit, including resistance to Mg^2+^ blockade and a remarkably long channel decay latency, might prove important for the integrative functions of the interneurons where this subunit is predominantly expressed. However, it should be noted that this receptor has a very low open channel probability of 1–4% and, as mentioned previously, lower Ca2+ permeability than the other GluN subtypes and is also considered a low-conductance NMDAR, which might imply that it is less effective at depolarizing the postsynaptic membrane and triggering downstream signalling events [39,140].

GluN2D-containing receptors play a role in both presynaptic and postsynaptic neurotransmission. Studies have reported GluN2D-containing NMDARs on interneurons in the hippocampus and neocortex where they play a key role in postsynaptic signalling [122,141]. GluN2D-containing receptors are thought to enable interneurons to synchronize and coordinate the firing of large groups of cortical pyramidal neurons. A recent study showed that tonic activation of these GluN2D-containing NMDARs on developing cortical interneurons is required for proper intrinsic excitability, dendritic arborization, GABAergic synaptogenesis and inhibitory tone onto excitatory pyramidal cells [141]. GluN2D-containing NMDARs have also been reported on the dendrites of neurons in the subthalamic nucleus, on dopaminergic projection neurons in the substantia nigra pars compacta and in the dorsal horn of the spinal cord, where they contribute to the modulation of the indirect pathway, mediate dopamine release to the striatum and play a role in pain perception, respectively [142,143]. GluN2D-containing NMDARs expressed at presynaptic sites are thought to play a modulatory role in the hippocampus and cerebellum. In the hippocampus, GluN2D, along with GluN2B and postsynaptic metabotropic glutamate receptors, have been shown to be critical for the induction of spike-time-dependent LTD [144]. This is a form of synaptic plasticity that occurs when the timing and order of spikes from the pre- and postsynaptic neurons leads to a long-lasting weakening of the synaptic connection between them, and is important for refining synapses and neural circuits both during development and learning and memory in adults [144,145]. In the cerebellum, presynaptic GluN2D-containing NMDARs help fine tune the release of GABA onto Purkinje cells [146].

Interestingly, it has been reported that the human *GRIN2D* gene contains four estrogen-responsive elements which are highly preserved in the rat, suggesting that the GluN2D subunit might be under neuroendocrine control [147]. In line with this, a study using ovariectomised rats found an upregulation of GluN2D mRNA in the hypothalamus following 17β-estradiol treatment [147]. Given the well-established links between fluctuating levels of 17β-estradiol and schizophrenia onset and symptom severity [148], it is intriguing to consider that one of the actions by which 17β-estradiol may exert its effects is via regulation of GluN2D and other NMDAR subunits [149].

Overall, considering its expression peaks during development, and that it is predominantly located on interneurons where its unique properties likely contribute to the integrative function of the interneurons, it is highly likely that the GluN2D receptor subunit plays an important role in the NMDAR hypothesis of schizophrenia.

## 5. Alterations to GluN2D in Schizophrenia

There is evidence to suggest that the GluN2D subunit is altered in schizophrenia. A study of approximately 200 Japanese people with schizophrenia found single nucleotide polymorphisms (SNPs) in the gene for the GluN2D receptor that might contribute to schizophrenia susceptibility [150]. They report that specific combinations of four SNPs within the *GRIN2D* gene were significantly associated with schizophrenia. These specific combinations were found in three pairs of SNPs: INT10SNP–EX13SNP2, EX13SNP2–EX13SNP3 and EX6SNP–EX13SNP2 [150]. A recent mutation-screening study also identified an ultra-rare, loss-of-function splice-site mutation (c.1412G>A) in the exonic region of the *GRIN2D* gene, which may lead to the creation of a truncated, nonfunctional GluN2D receptor, thereby contributing to schizophrenia risk [151]. This study additionally found four missense mutations in schizophrenia patients in the *GRIN2D* gene, and although the actual functional impact of these amino acid substitutions was not examined, in silico analysis classified each of the four variants as disease-causing based on their predicted effect on protein function [151]. An in situ hybridization study on post-mortem human tissue reported a 53% increase in the proportion of GluN2D mRNA expression in the PFC of people with schizophrenia (see Table 1) [128]. This increase in the proportion of GluN2D expression was not seen in anti-psychotic-treated control brains, proving that the change was specific to schizophrenia and could not be attributed to treatment with anti-psychotics alone [128]. However, in another in situ hybridization study, no change in the mRNA levels of the GluN2D subunit was observed in the dlPFC of people with schizophrenia [152]. Furthermore, in a post-mortem Western blot study, no change in the protein expression of the GluN2D subunit was observed in the dlPFC and anterior cingulate cortex in elderly people with schizophrenia when compared with a control group [153]. In one human post-mortem study, laser microdissection was used to isolate a population of glutamatergic relay neurons and another mixed glial and GABAergic interneuron population (including PV+ interneurons) from the medial dorsal thalamus of people with schizophrenia and controls [154]. This study revealed decreased expression of the GluN2D transcript in glutamatergic relay neurons in the medial dorsal thalamus, whilst no changes were observed in the mixed glia and interneuron population in schizophrenia [154]. These relay neurons project to the PFC, suggesting that alterations to GluN2D in schizophrenia might contribute to the disruption of the thalamocortical circuit thought to be involved in the attention and sensory processing deficits and negative symptoms seen in schizophrenia [155,156,157]. Another study of a small sample size (n = 19) found that GluN2D gene expression was increased in the right cerebellum of post-mortem human schizophrenia brains compared to controls [158]. However, this study included people with schizophrenia who had been taking anti-psychotic medications chronically, which could have contributed to any differences observed compared to controls. These discrepant results highlight the need for further studies examining the region- and cell-specific expression and function of the GluN2D subunit in the brains of people with schizophrenia. In the following sections, the expected consequence of mutations in GRIN2D and changes in GluN2D expression are discussed.

## 6. Consequences of Loss of GluN2D Function

Strategies to explore the influence of GluN2D function include using animal models and pharmacological tools to study behavioural and physiological measures which have relevance to schizophrenia.

### 6.1. Genetic Models

GluN2D-knockout (KO) mice are viable, reproduce and grow normally, and have no overt changes in neuronal histology [159]. Moreover, mRNA levels of the other NMDAR subunits are unaffected in these mice [159]. These mice, however, exhibit unique behavioural phenotypes, including diminished spontaneous motor movements in open-field tests (Table 2) [159,160]. GluN2D-KO mice also display deficits in spatial learning and memory, as well as impaired contextual fear memory, but show no deficits in the novel object recognition task [137,161,162]. Most studies report no abnormalities in motor function as measured by the rotarod test, nor any differences in anxiety when compared with WT mice during the light–dark compartment test and elevated maze test [159,163]. However, Miyamoto et al. report reduced sensitivity to stress induced by the elevated-plus maze, the light–dark compartment test and forced swim tests in these KO mice [160]. The discrepancies in these findings might be due to the differences in the way the tests were performed, which could be sensitive to subtly different anxiety- or fear-related behaviours. This study also reported a downregulation of dopamine, serotonin, norepinephrine and their metabolites in the hippocampus and striatum of GluN2D-KO mice [160]. GluN2D-KO mice are reported to have a normal prepulse inhibition response, which is known to be deficient in schizophrenia [137,164]. This suggests that they might not model all aspects of schizophrenia. Interestingly, the hyperlocomotion-inducing effects of PCP and ketamine are reduced in GluN2D-KO mice, indicating that the GluN2D subunit plays an important role in mediating the effects of these drugs [137,163,165]. Additionally, the motor impairments, locomotor sensitization and increase in extracellular dopamine seen following PCP or ketamine administration in WT mice is not seen in GluN2D-KO mice [163,165,166]. These data suggest that the GluN2D subunit mediates the hyperdopaminergic-inducing effects of the NMDAR antagonists, PCP and ketamine. Autoradiography studies revealed that the increase in neuronal activity induced by sub-anaesthetic ketamine in the medial prefrontal cortex, entorhinal cortex, presubiculum and caudate putamen in WT mice is not seen in GluN2D-KO mice [98,137,167].

GluN2D-KO mice were reported to have similar basal neural oscillatory power between frequency ranges 30 and 200 Hz when compared with WT mice [168]. However, whilst the administration of the NMDAR antagonists MK-801, ketamine and memantine increased oscillatory power in WT mice, they had very little effect on GluN2D-KO mice, especially at the high gamma frequency range (65–140 Hz) [168]. Similarly, Sapkota et al. reported that ketamine increased gamma frequency (30–140 Hz) oscillatory power in WT mice but elicited a much less pronounced increase in GluN2D-KO mice, which was more apparent at the high frequency (>60 Hz) range [137]. This suggests a role for the GluN2D subunit in modulating the high-frequency neural network oscillations induced by NMDAR antagonists. The study by Sapkota et al. also found reduced PV cell density in the substantia nigra and basolateral/lateral amygdala and a trend towards a reduction in the mPFC and hippocampus in GluN2D-KO mice compared with WT mice [137].

These studies suggest the GluN2D subunit expressing NMDA receptors are critical in mediating many of the effects of NMDAR antagonists, like ketamine, on the behaviour and electrophysiology that is relevant to schizophrenia, and thus may, in turn, suggest a role for the GluN2D subunit in the development of schizophrenia. There is evidence to suggest that GluN2D-containing NMDARs can contribute to neuronal networks that underlie cognition and which are found to be disrupted in schizophrenia. GluN2D-mediated signalling could be promising as a potential therapeutic target for specific symptoms observed in individuals with schizophrenia.

### 6.2. Pharmacological Manipulations

Although currently, to the best of our knowledge, there are no GluN2D-selective drugs available, competitive antagonists with 3–10-fold higher selectivity for GluN2C/GluN2D compared with GluN2A/GluN2B-containing NMDARs have been developed. One such compound, (2R*,3S*)-1-(phenanthrenyl-2-carbonyl)piperazine-2,3-dicarboxylic acid (PPDA), resulted in more potent inhibition of LTD than LTP in rat hippocampal slices, suggesting a role for the GluN2D (and GluN2C) NMDAR subunits in hippocampal LTD [169]. Another study using PPDA showed that GluN2D-containing NMDARs also contribute to extrasynaptic currents in rat CA1 neurons [170]. A recent study used an analogue of PPDA, (2R*,3S*)-1-(9-bromophenanthrene-3-carbonyl)piperazine-2,3-dicarboxylic acid (UBP145), in combination with GluN2D-KO mice, to show that GluN2D-containing NMDARs play a role in short-term potentiation (STP) as well as LTP in the mouse hippocampus [171]. This study reported that UBP145 partially inhibited LTP and the slow component of STP in WT but not GluN2D-KO mice [171]. Whilst the function of STP is lesser known than LTP, it has been hypothesized that it might play a role in forms of short-term memory, such as working memory which is known to be disrupted in schizophrenia [172]. These data together suggest that GluN2D-containing NMDARs contribute to synaptic plasticity and thus cognitive processes in complex ways. Therefore, it is possible that dysregulation of these receptors is involved in the cognitive dysfunction seen in schizophrenia.

## 7. How Might Alterations to the GluN2D Subunit Contribute to Schizophrenia?

Several characteristics of the GluN2D subunit, including peak expression early in development, its localization to PV+ interneurons in the cortex and hippocampus and, additionally, reports of alterations to this subunit in post-mortem tissue from people with schizophrenia, and evidence of *GRIN2D* being a schizophrenia candidate gene, suggest that it may be involved in schizophrenia pathology. Below, we propose a mechanism by which dysfunction of the GluN2D receptor could contribute to schizophrenia.

### 7.1. GluN2D Subunit and Parvalbumin-Positive GABAergic Interneurons

As previously discussed, inhibitory interneurons have been identified as the key locus or point of convergence of the glutamatergic, GABAergic and dopaminergic hypotheses of schizophrenia and are also implicated in the cognitive deficits seen in schizophrenia. Interestingly, in situ hybridization, electrophysiology and immunohistochemistry studies have revealed that GluN2D-containing NMDARs are specifically enriched in the PV-expressing subclass of interneurons in the hippocampus and PFC, two regions that underlie learning and memory function (refer to Section 4.1) [120,121,122,141]. This makes the GluN2D subunit particularly intriguing in the context of schizophrenia as several studies have shown that the hypofunction of NMDARs at fast-spiking PV-containing interneurons is sufficient to produce schizophrenia-like symptoms, including cognitive dysfunction [84,173,174,175].

Despite there being more than 20 different classes of GABAergic interneurons [176], it is the interneurons containing the calcium-binding protein, PV, that have been proposed to be especially important in schizophrenia [177,178]. Not only are PV-containing interneurons crucial for regulating cortical inhibition via the pyramidal neurons they innervate, but also for the generation of synchronous gamma-frequency oscillations [178,179,180,181,182,183]. Gamma oscillations are synchronous electrophysiological brain rhythms in the gamma frequency range (30–80 Hz) that are crucial for information processing and appropriate cortical function and underpin a wide range of cognitive processes, including those disrupted in schizophrenia like working memory [184,185,186]. Abnormal gamma-frequency synchrony is a major pathological characteristic of schizophrenia and underlies cognitive deficits [186]. For example, a recent study found lower-amplitude gamma oscillations in people with schizophrenia while they were performing a working memory task [187]. Similarly, in another cohort of people with schizophrenia, gamma-band activity was reduced during the sensory processing state during an auditory task, but baseline gamma power during the resting state was increased when compared with healthy controls [188,189]. Additionally, NMDA antagonists like ketamine and PCP also induce these same gamma oscillatory disturbances in healthy humans and rodent models of schizophrenia [95,190,191,192,193,194,195]. Thus, it is possible that dysfunction of the GluN2D subunit, which is especially enriched in PV cells in the cortex and hippocampus, might contribute to the abnormal gamma oscillations and the associated cognitive deficits seen in schizophrenia. Indeed, studies have found that genetic ablation of the GluN2D subunit from mice significantly reduces NMDAR antagonist-induced high-frequency gamma oscillations [137,168]. Alterations of the GluN2D subunit could affect the ability of PV cells to mediate the inhibition of excitatory pyramidal cells in a synchronised manner, disrupting the E/I balance and resulting in abnormal neural oscillations. Indeed, a recent study has found that conditional deletion of the GluN2D subunit from PV interneurons resulted in hyperexcitation of the medial PFC and disruptions to the feedforward inhibitory circuit [196]. Behaviourally, these mice exhibited hyperlocomotion, increased anxiety-like behaviour and impaired short-term memory and cognitive flexibility [196]. This study also found that GluN2D ablation from PV interneurons resulted in a downregulation of genes involved in GABAergic and dopaminergic synapse function such as *GAD67* and *TH*, as well as schizophrenia susceptibility genes such as *Disc1*, *ErbB4* and their downstream targets [196], thus suggesting a critical role for the GluN2D subunit in PV cells in modulating schizophrenia-relevant changes in neural circuitry, signalling and behaviours. The selective expression of the GluN2D subunit in PV+ interneurons suggests it may be a potential therapeutic target that could reverse interneuronal hypofunction and the currently untreated cognitive impairments that result from these deficits. Positive modulation of the GluN2D-subunit-containing NMDA receptor indeed increases the firing rates and restores the GABAergic network stability and reversed working memory deficits in a mouse model of schizophrenia symptoms [121,122,197]. However, as the GluN2D subunit is so critical to development, it may prove a challenge to reverse any deficits. A recent study showed that tonic activation of GluN2C/GluN2D-containing receptors during development is needed for proper cortical PV interneuron morphological maturation and complexity, circuit integration and maintaining proper inhibitory tone onto excitatory pyramidal cells [141]. The study found that blockade of GluN2C/GluN2D-containing receptors between post-natal days 7 and 9 was sufficient to cause long-lasting cortical inhibitory network deficits, as seen in schizophrenia [141].

Cortical pyramidal neurons innervated by PV cells stimulate dopaminergic neurons in the midbrain which project to the associative striatum. The associative striatum includes the rostral and dorsal part of the caudate nuclei and is implicated in the pathophysiology of schizophrenia [198,199,200,201]. The associative striatum is rich in dopamine receptors and dopamine afferents and receptors and is thus thought to be the primary site of action of antipsychotics. As such, any disruption to the GluN2D-containing NMDARs on PV cells could also indirectly affect midbrain dopaminergic neurons and lead to enhanced dopamine release in the striatum which has been linked to the positive symptoms of schizophrenia [202]. Following treatment with ketamine, Yamamoto et al. found an increase in locomotor activity and nitric oxide (NO) synthesis in the dendrites of medium spiny neurons in the dorsal striatum and PFC in WT but not GluN2D-KO mice [166]. Postsynaptic neuronal NO synthesis is functionally coupled to the stimulation of NMDARs. The failure of ketamine to induce an increase in striatal NO synthesis in GluN2D-KO mice provides support for the role of GluN2D-containing receptors in the corticostriatal neuronal circuit. Similarly, PCP induced an increase in the number of Fos-positive (marker of neuronal activity) cells in the striatum, prefrontal cortex, thalamus and subthalamic nuclei of WT but not GluN2D-KO mice [165]. This suggests that the frontal cortex–dorsal striatum pathway is likely activated by the inhibition of GluN2D-containing NMDARs.

### 7.2. GluN2D Subunit and Dopaminergic Neurons

Multiple studies have found that the GluN2D subunit forms functional NMDAR channels in the substantia nigra pars compacta dopaminergic neurons [131,203,204,205]. The substantia nigra plays an essential role in modulating motor movement and reward functions. Interestingly, GluN2D-KO mice have a hypolocomotor phenotype and the hyperlocomotor effects of PCP and ketamine are reduced in GluN2D-KO mice [137,163,166]. The amount of dopamine release in the forebrain following PCP treatment is also reduced in GluN2D-KO mice [163]. This suggests that GluN2D-containing NMDARs might play either a direct or indirect role in modulating dopaminergic function and, consequently, locomotor activity. In people with schizophrenia, there are reports of long-term deficits in basic motor function and control [206,207,208]. Thus, it is possible that dysfunction of the GluN2D subunit could affect burst-firing in these dopaminergic neurons, disrupting their function in the nigrostriatal circuitry, which is hypothesised to underlie motor symptoms in schizophrenia [209]. Impairments in reward processing have also been reported in people with schizophrenia and may underlie some of the negative symptoms in schizophrenia, such as anhedonia and a lack of motivation [210,211]. A study by White et al. reported that glutamate in the substantia nigra plays a role in reward processing and found that glutamatergic dysfunction in the substantia nigra could contribute to reward-processing deficits [212]. Thus, any alterations to GluN2D-containing NMDARs in the substantia nigra might also play a role in reward-processing deficits. Indeed, Yamamoto et al. reported that ablation of the GluN2D subunit in mice resulted in an anhedonic state, indicated by a reduction in sucrose preference [166]. Additionally, loss of GluN2D-containing NMDARs resulted in anxiety- and depressive-like behaviours in mice which was linked to disruptions to the modulation of neural activity by GluN2D-containing NMDARs in the bed nucleus of the stria terminalis [213]. Together, these studies suggest a role for GluN2D-containing NMDARs in mediating emotional behaviours that are known to be affected in schizophrenia.

## 8. Limitations

As this is a narrative rather than a systematic review, there is potential for bias in the selection of publications which may, in turn, bias the conclusions of this review. A scoping review with clearly developed, predetermined inclusion and exclusion criteria and keywords might have led to more replicable and verifiable conclusions. Furthermore, due to the heterogeneity of the available literature on the GluN2D subunit, we are unable to make a conclusive statement on the role of the GluN2D subunit in schizophrenia. Instead, our review of the literature suggests that further research needs to be undertaken to fully understand the contribution of the GluN2D subunit to this disorder. A major contributor to the observed heterogeneity may be that the changes in GluN2D in schizophrenia are cell-specific. Here, recent advances in single-cell and/or single-nuclei RNA sequencing will enable a more detailed understanding of the role of GluN2D in schizophrenia. Additionally, we were not able to adequately investigate any potential sex-specific differences in the expression of the GluN2D subunit or its function or effects in humans or animal models as sex was not always included as a variable in the studies examined here. This stresses the importance of future work to always include sex as a variable.

## 9. Conclusions

Precipitating factors, including any combination of genetic predisposition and environmental factors like maternal infection or obstetric complications, can lead to NMDAR hypofunction disproportionately at fast-spiking PV-containing interneurons during development. This results in pathological phenotypes including impaired oscillatory activity and neuronal synchrony, cortical disinhibition and dopaminergic dysfunction, ultimately giving rise to the various symptoms of schizophrenia. Disruption of the GluN2D subunit and alteration to GluN2D neurotransmission could be a molecular pathway contributing to the symptomatology of schizophrenia. This is of importance as it may provide new insights into the aetiology of this disorder and might even lead to the development of novel drugs for the treatment of specific schizophrenia symptoms, including cognitive dysfunction.

## Figures and Tables

**Figure 1 ijms-24-11835-f001:**
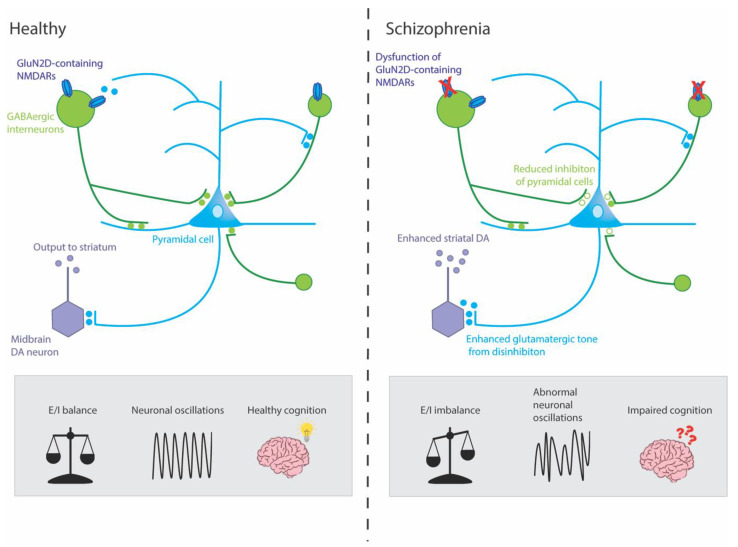
Mechanisms by which dysfunction of GluN2D-containing NMDARs could potentially result in symptoms of schizophrenia. This simplified circuit shows that, in healthy individuals, GluN2D-containing NMDARs stimulate GABAergic interneurons on which they are expressed, which, in turn, mediate inhibition and coordinate the synchronized firing of networks of excitatory pyramidal neurons. These pyramidal neurons, in turn, stimulate dopaminergic (DA) neurons in the midbrain (via the ventral striatum and ventral pallidum, not shown) which project to the associative striatum. In healthy individuals, this circuit is under homeostatic balance, leading to gamma oscillations and healthy cognition. In schizophrenia, hypofunction of the GluN2D-containing NMDARs could lead to GABAergic interneurons increasing excitation of pyramidal neurons by reducing inhibition (disinhibition) onto pyramidal neurons. The resulting aberrant increase in cortical excitation would lead to abnormal neuronal oscillations and impaired cognition. Moreover, this hyperactivity might lead to an overdrive in midbrain DA neurons and enhanced striatal DA, which has been linked to the positive symptoms of schizophrenia.

**Table 1 ijms-24-11835-t001:** Summary of post-mortem studies of GluN2D NMDAR subunit expression in schizophrenia.

Reference	Outcome Measure	Brain Region	Direction of Change
[128]	mRNA	Prefrontal cortex and cerebellum	↑ in PFC ↔ in cerebellum
[152]	Receptor binding and mRNA	Dorsolateral prefrontal cortex	↔
[153]	Protein	Dorsolateral prefrontal cortex and anterior cingulate cortex	↔
[154]	mRNA	Medial dorsal thalamus	↓ in glutamatergic relay neurons ↔ in mixed glial-interneuronal cells
[158]	Receptor binding and mRNA	Cerebellum	↑ mRNA in the right cerebellum ↔ mRNA in left cerebellum and receptor binding

PFC = prefrontal cortex, mRNA = messenger ribonucleic acid. ↑ = increased expression, ↓ = decreased expression, ↔ = no change.

**Table 2 ijms-24-11835-t002:** Summary of schizophrenia-relevant behavioural phenotypes in GluN2D-knockout mice.

Reference	Sex	Pharmacological Manipulation and Dose	Behavioural Domain Examined	Behavioural Tests Used	Main Findings
[159]	Not specified	N/A	Locomotion, anxiety, novelty preference	Open field test, novelty preference test, light–dark compartment test, elevated plus-maze	↓ spontaneous locomotion, ↓ novelty preference, ↔ change in anxiety
[160]	Not specified	N/A	Locomotion, anxiety	Open field test, light–dark compartment test, elevated plus-maze, forced swim test	↓ spontaneous locomotion, ↓ anxiety
[163]	Male	Acute and chronic PCP (3 mg/kg)	Locomotion	Open field test	↓ PCP-induced hyperlocomotion
[161]	Male	N/A	Locomotion, contextual fear memory, spatial memory	Open field test, fear conditioning test, Y-maze	↓ spontaneous locomotion, ↓ contextual fear memory, ↓ spatial memory
[165]	Male and female	Acute PCP (3 mg/kg)	Motor function	Rotarod	↓ PCP-induced motor impairment
[137]	Male	Acute ketamine (30 mg/kg)	Locomotion, spatial memory	Open field test, Morris water maze	↓ ketamine-induced hyperlocomotion, ↓ spatial memory acquisition
[166]	Male	Subchronic ketamine (25 mg/kg)	Locomotion	Open field test	↓ ketamine-induced hyperlocomotion
[162]	Not specified	Acute (*RS*)-ketamine (10 or 20 mg/kg), (*R*)-ketamine (10 or 20 mg/kg), (*S*)-ketamine (10 or 20 mg/kg)	Novel object recognition task	Novel object recognition task	↓ (*R*)-ketamine-induced novel object recognition deficits

N/A = not applicable, PCP = phencyclidine. ↓ = decreased, ↔ = no change.

## Data Availability

No new data were created for this review.

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
