# Peer review of "Potential Roles for the GluN2D NMDA Receptor Subunit in Schizophrenia"

_ijms, 2023, doi:10.3390/ijms241411835_

Round 1

Reviewer 1 Report

           The studies described in the review article "Potential roles for the GluN2D NMDA receptor subunit in schizophrenia" by Vinnakota et al. reviewed the available evidence (genetic and pharmacological) on the role of GluN2D subunit of NMDA receptors in schizophrenia.

The following are my suggestions to improve the article:

1.    Abstract section, Line 12-13: Please rephrase the sentence for better clarity.

2.    Line 15- What do the authors mean by “Schizophrenia-relevant characteristics”?

3.    Line 16- Expand GluN2D

4.    Line 36- Cognitive deficits

5.    Line 61-64: Please cite original articles corresponding to studies showing the association of the environmental factor mentioned with schizophrenia.

Further, wherever possible, the authors are encouraged to cite original research articles rather than review articles in the current manuscript.

6.    Line 100: Please cite research articles showing the link of dopamine to positive symptoms only. Also, the other study mentioned here is related to ketamine only.

7.    Line 102-The Ref 33 cited by the authors has not profiled the levels of dopamine receptors in the frontal cortex. This study has used an image from brain-atlas.org to show the distribution of Dopamine D2 receptors in various brain areas. The authors should be cautious in citing the right reference in the manuscript.

8.    Page 3 first paragraph: At the beginning of this paragraph, the authors are saying that dopaminergic dysfunction is predominantly related to positive symptoms. And in the concluding section, the authors are associating changes in the Glutamatergic system with DAergic dysfunction, E/I imbalance, and all three symptoms. Further, the authors mention dopamine receptors in the PFC, but none of the quoted studies say about the NMDA antagonist-mediated dysregulation of dopamine release in the PFC. In my opinion, the authors need to rewrite this entire paragraph.

9.    The authors should move sections 3 and 4 (information about glutamate, NMDA receptors, their subunit compositions, corresponding genes, and the agonist and co-agonists) before explaining studies indicating NMDA in schizophrenia (section 2). This will be very helpful for the readers to follow the review.

10.               Line 137-139: Please rephrase this statement for clarity.

11.               Line 150-154: Very long sentence and not easy to understand.

12.               Line 352-354: Please rephrase this statement.

13.               Line 392: What is associative striatum?

14.               Line 393-395: The sentence seems incomplete.

15.               Line 407: What do the authors mean by “NMDAR-mediated input”?

16.               In the NMDAR hypofunction on GABAergic interneurons section, the authors need to mention the specific brain areas they are referring to as these neurons are expressed in multiple brain areas.

17.               Line 416-418: The authors are citing a review article from 2005 to suggest the role of PV-INs in schizophrenia. Please quote recent articles as well.

Re-phrasing of a few sentences is required. 

Author Response

Please check the attached cover letter

Reviewer 2 Report

This review article by Vinnakota and colleagues states its aim as being discussing potential roles for the GluN2D subunit of the NMDA receptor in schizophrenia. Individually, most sections of the manuscript are well reviewed and written, however the way these elements are arranged is far from ideal. It rather feels like two reviews - one of the authors’ vision of the NMDA receptor hypothesis of schizophrenia, and another on the detailed biology of the GluN2D subunit – with these two reviews having been mixed together.

Notably, the authors open with an overview of schizophrenia (1), which builds into the NMDA receptor hypothesis of schizophrenia (2). Only after doing this, however, do they have a section describing what the NMDA receptors are (3-start of 4). They then go into a detailed discussion of the biology of their subunit of interest (4.1, 4.2, 5, 6.2, 7), which is interrupted by a detailed return to the NMDA receptor hypothesis (6.1).  Its not that there is no logic at all to this, I can understand what the authors were aiming for, but it comes across as convoluted and disorientating.

The second major issue is that the two aspects of the review (the NMDA receptor hypothesis of schizophrenia and GluN2D biology) are tied together by a short section on evidence connecting the subunit to schizophrenia – however this section is very short, and the evidence described is fairly sparse. There are several studies potentially implying a role for GluN2D, based on diverse approaches, but a distinct lack of replication.  This does not mean that that authors cannot write about GluN2D and schizophrenia – discussion stimulates research – but it makes for a weak basis to connect grand discussions of the bases of schizophrenia with the biology of this specific receptor. This is epitomized by section 6 whose title suggests that it will present a theoretical framework for how the subunit functions in schizophrenia, but instead consists of a long discussion of the NMDA receptor hypothesis, detailed descriptions of GluN2D based experiments, and then only one short paragraph to tie them together. The end result is that, even the authors are specific in describing the data linking GluN2D to schizophrenia, the implied connection ends up being greatly oversold.

My strong advice to the authors would therefore be to focus on only one of these aspects and cut or greatly deemphasize the other: either give a detailed review of GluN2D and its potential involvement with schizophrenia, but with the grander ideas of schizophrenia biology significantly cut back, or write a review of the NMDA receptor hypothesis of schizophrenia more generally.

Author Response

Please check the attached cover letter

Reviewer 3 Report

The topic is actual, because schizophrenia is a severe, debilitating, chronic neuropsychiatric disorder. The causes of schizophrenia are not fully understood but are thought to be multifactorial, involving a complex interplay between multiple genetic variants and environmental factors. The authors propose a mechanism by which dysfunction of the GluN2D receptor could contribute to schizophrenia. In this article, the authors show that disruption of the GluN2D subunit and alteration of GluN2D neurotransmission may be a molecular pathway contributing to the symptomatology of schizophrenia. This is important as it may provide new insight into the etiology of this disorder and may provide a biochemical basis for the development of new drugs to treat the specific symptoms of schizophrenia, including cognitive dysfunction. The article is written in good language, well-structured and interesting to read.

Author Response

Please check the attached cover letter

Reviewer 4 Report

The manuscript is devoted to the topical topic of biological psychiatry - the glutamatergic hypothesis of schizophrenia. This hypothesis has served as the basis for a large number of studies on animal models, especially over the past 20 years. The results of recent studies indicate that the glutamatergic hypothesis may lead to a revision of the concept and existing approaches to the treatment of schizophrenia, which would be impossible on the basis of the dopaminergic hypothesis alone.

In the manuscript, the authors consider evidence of changes in the components of glutamate chains and signaling pathways using the example of the GluN2D NMDA receptor subunit.

The manuscript is well structured and easy to read, but it looks like a chapter in a monograph, but not a scoping  or narrative review. The introductory part is unnecessarily detailed and contains well-known encyclopedic data. In the main part there are no tables reflecting an overview of the current results of preclinical (cell culture, animal model) and clinical (patients with schizophrenia, autopsy material) studies of the role of the GluN2D NMDA receptor subunit confirming (positive results) or rejecting (negative results) the glutamatergic hypothesis of schizophrenia.

Unfortunately, the authors analyzed outdated data - 120 publications out of 181 cited by the authors were published more than 10 years ago. Thus, the authors of the submitted manuscript do not adhere to the methodology of preparing a scoping review and the guidelines for authors posted on the MDPI website.

The scoping review should reflect the size, characteristics or coverage of available publications in the field of interest over the past 5 years (maximum, 10 years) in order to help identify research gaps and new needs. In 2018, the PRISMA extension for scoping reviews was published.

If the authors wanted to prepare a narrative review, then in this case they should evaluate, criticize and summarize the available research on the topic over the past 5 years (maximum, 10 years). The narrative review is much less systematic and meticulous, although it is based on evidence. However, this type of review is not always considered extremely useful in terms of the scientific evidence it provides, since the authors of such a review are much more prone to biased selection.

Any of the above reviews is a great undertaking, and very useful.

So, the manuscript needs a serious revision: a) the authors need to make a choice between scoping and narrative review (scoping review is preferable to increase the interest of readers of the journal); b) decide on a strategy for subjective search and comparison of previously published data; c) if the authors choose a descriptive review, then a section on limitations should be added to the manuscript, where the subjective inclusion and exclusion criteria for the primary study should be noted, as well as potential bias in the selection.

Finally, it is unclear how this review differs from the previously published ones. Perhaps the authors will present their new view on the glutamatergic hypothesis of schizophrenia under consideration.

Author Response

Please check the attached cover letter

Round 2

Reviewer 1 Report

The following are suggestions to further improve the article:

1. Line 1: Expand NMDA

2. Line 14- "drugs" should be replaced with "compounds".

3. Line 82-85: Lengthy sentence.

4. Line 84-85: Please check for grammar issue (suggest).

5. Transition from NMRDR hypofunction section to GluN2D is not smooth. The authors should discuss briefly about about what is know about other subunits in schizophrenia and why GluN2D is of their interest.

6. Line 434- Please rephrase the statement.

Minor editing of the English language is required

Author Response

Please check the attached cover letter

Reviewer 2 Report

The authors have addressed all of my major concerns, and I am happy to recommend the manuscript for publication.

Author Response

Please check the attached cover letter

Reviewer 4 Report

The authors have significantly improved the manuscript.

It needs minor correction:

1) Please pay attention to the abbreviations. For example, the section name uses "CNS", further in the text (for example, line 351) - "central nervous system". If the abbreviation is used four or more times, then it is better not to use it.

2) Tables 1 and 2 have abbreviations. Please add a Note under the tables and explain all the abbreviations used.

3) The number of cited publications older than 5 years is still very high, and older than 10 years - about 50%. However, I leave this comment for the Editor's consideration.

Author Response

Please check the attached cover letter
